# Muscle-like Scaffolds for Biomechanical Stimulation in a Custom-Built Bioreactor

**DOI:** 10.3390/polym14245427

**Published:** 2022-12-11

**Authors:** Laura Rojas-Rojas, María Laura Espinoza-Álvarez, Silvia Castro-Piedra, Andrea Ulloa-Fernández, Walter Vargas-Segura, Teodolito Guillén-Girón

**Affiliations:** 1Materials Science and Engineering School, Instituto Tecnológico de Costa Rica, Cartago 30101, Costa Rica; 2Physics School, Instituto Tecnológico de Costa Rica, Cartago 30101, Costa Rica; 3Biology School, Instituto Tecnológico de Costa Rica, Cartago 30101, Costa Rica

**Keywords:** tissue engineering, myoblasts, viability, cell adhesion, bioreactor

## Abstract

Tissue engineering aims to develop in-vitro substitutes of native tissues. One approach of tissue engineering relies on using bioreactors combined with biomimetic scaffolds to produce study models or in-vitro substitutes. Bioreactors provide control over environmental parameters, place and hold a scaffold under desired characteristics, and apply mechanical stimulation to scaffolds. Polymers are often used for fabricating tissue-engineering scaffolds. In this study, polycaprolactone (PCL) collagen-coated microfilament scaffolds were cell-seeded with C2C12 myoblasts; then, these were grown inside a custom-built bioreactor. Cell attachment and proliferation on the scaffolds were investigated. A loading pattern was used for mechanical stimulation of the cell-seeded scaffolds. Results showed that the microfilaments provided a suitable scaffold for myoblast anchorage and that the custom-built bioreactor provided a qualified environment for the survival of the myoblasts on the polymeric scaffold. This PCL-based microfilament scaffold located inside the bioreactor proved to be a promising structure for the study of skeletal muscle models and can be used for mechanical stimulation studies in tissue engineering applications.

## 1. Introduction

Tissue engineering remains the focus of scientific research due to its applications in biomedical areas and disease modeling. This field aims to reconstruct tissues and organs through the use of cells, scaffolds, and environmental factors [1,2]. These three fields should be integrated to produce optimal tissue functionality and promote cell development [3]. Myoblasts are able to regenerate and reproduce, however, they require a surface to anchor in order to support tissue formation [4,5]. An appropriate cell matrix or scaffold plays a crucial role in cell development [4,6]. Scaffolds for tissue engineering can be three-dimensional structures that provide an adhesion surface for cell proliferation and differentiation [7,8], for instance, important biological processes like gene expression, cellular mobilization, and apoptosis can differ significantly between two-dimensional cultures and three-dimensional cell cultures [9]. Scaffolds should comply with design parameters that mimic the mechanical, physical, and biological properties of native muscular tissue [10,11]. Hence, when manufacturing a scaffold for musculoskeletal tissue engineering applications, it must mimic the skeletal muscle physiology and its mechanical properties [12,13]. In addition, a muscular tissue engineering scaffold should aim to imitate the collective behavior of a muscle and have elastic properties [14].

Polycaprolactone (PCL) is a biomaterial widely used for scaffolds in tissue engineering. PCL is a synthetic polymer, biocompatible and bioabsorbable [15,16]. According to Siddiqui et al. [17], PCL stands out among other biomaterials for its capacity to be molded into different shapes. Another very important feature of this material is that it has been approved by the Food and Drug Administration (FDA) for medical purposes [16,18]. PCL scaffolds can be designed with different architectures and can be fabricated by additive manufacturing and electrospinning [19,20,21]. Mats, meshes, or porous structures can be fabricated from this material [22,23,24,25]. Ghobeira et al. [26] reported PCL scaffolds fabricated by a modified fused deposition modeling (FDM) method with random or aligned fiber configurations. Other techniques rely on the thermoplastic properties of PCL, such as extrusion and melting-drawing techniques [27,28]. These techniques are used to produce individual polymeric fibers—bioprinting of fibers to create muscle-like structures can be used [29].

Collagen is one of the principal components of the extracellular matrix and can be naturally found in tendons, bones, and skin [30,31]. This natural polymer is commonly used as a scaffolding material or coating material, as it promotes cell adhesion and proliferation [1,30]. Collagen used as a coating material has also been related to an enhancement of the biocompatibility of a base material [24,32]. Other options include covering a surface with hydroxyapatite (HA) to create a biocompatible layer. Egorov et al. [33] present different coating techniques to enhance biocompatibility.

Bioreactors represent a system capable of initiating, maintaining, and directing cell growth and tissue formation [34]. These systems offer dynamic culture conditions that are able to recreate the physiological environment of native tissue because variables like temperature, pH, medium flow rate, nutrient supply, and gas can be controlled to match the requirements of the cell culture [35,36]. Design and construction of bioreactors allow the application of mechanical stimuli such as compression and tension to cell-seeded scaffolds to modify their characteristics and get the desired outcome [9]. Mechanical stimulation cues are especially relevant to muscle cells, which are intrinsically mechanical sensitive [37]. Their actin cytoskeleton is directly related to the cell’s mechanical response and its properties [38]. Different mechanical loading schemes can be used to elongate or compress scaffolds, this way direct cell reactions may occur [29,39,40].

This study focuses on studying a cell-seeded musculoskeletal scaffold inside a custom-made biomechanical stimulation system. A microfilament scaffold was fabricated to mimic the three-dimensional native configuration of skeletal muscle tissue and it was coated with collagen to enhance its biocompatibility. The microfilament was cell-seeded and placed inside the in-vitro system while another cell-seeded scaffold was left as a control. The test consisted in comparing myoblast cell proliferation on a microfilament scaffold inside the study system to that of a microfilament scaffold inside a cell growth incubator. The results of this test provided insights into whether the biomechanical stimulation system was adequate towards cell growth, whether it complied with aseptic conditions, and whether the environmental parameters, temperature, and CO_2_, were appropriate. Moreover, a preliminary stimulation test was made on the microfilament scaffold. This work provided information for future mechanical stimulation assays on cell-seeded scaffolds by using myoblast cells combined with a biomimetic muscular scaffold inside the custom-made study system.

## 2. Materials and Methods

### 2.1. Materials and Reagents

Polycaprolactone, PCL (Sigma-Aldrich, Mn 80,000 g/mol, Milwaukee, WI, USA) pellets were used to fabricate PCL microfilaments for the scaffolds and Collagen I (Sigma-Aldrich, Cat No. 125-50, lot 404, St. Louis, MO, USA) was used to coat the scaffolds. The cells used through this work were murine myoblasts C2C12 cell-line (CRL-1772^TM^) (American Type Culture Collection, ATCC, Manassas, VA, USA).

Cell proliferation assays were performed in cell growth medium which consisted of Gibco Dulbecco’s Modified Eagle Medium (DMEM-Grand Island, NY, USA) supplemented with 1% glutamine (GIBCO, Grand Island, NY, USA), 1% penicillin-streptomycin (P/S) (GIBCO, Grand Island, NY, USA), and 10% Fetal Bovine Serum (FBS) (GIBCO, Grand Island, NY, USA). Triton^TM^ X-100 (Lot LBC2688V SIGMA, Saint Louis, MO, USA) was used as cell lysis reagent. Trypsin-EDTA 0.25% and Phosphate Buffer Saline (PBS-SIGMA, cat no. P4417, lot SLCC1522, Saint Louis, MO, USA) were used to detach/release cells from culture vessel. Paraformaldehyde (PFA) (cat. No. O4042-500, Thermo Fisher Scientific^TM^, Waltham, MA, US) was used to fix the cells.

To analyze cell viability, AlamarBlue^TM^ Cell Viability Reagent (Invitrogen^TM^, Thermo Fisher, cat. No. DAL1100, Eugene, OR, USA) was used. For nuclei and cytoskeleton staining, 4′,6-diamidino-2-phenylindole (DAPI) (cat. No. sc-3598, Thermo Fisher Scientific) plus Alexa Fluor^TM^ 488 phalloidin (cat. No. A12379, Thermo Fisher Scientific) were used. Bovine Serum Albumin (BSA) (cat. No. BP9700-100, Thermo Fisher Scientific) was used to reduce nonspecific background staining.

All reagents for cell growth had analytical grade quality and proliferation test methods were assessed under aseptic techniques and sterile conditions.

### 2.2. Design and Fabrication of the In-Vitro Study System

The system for mechanical stimulation of a cell-seeded scaffold was made from 316 L stainless steel. It consisted of the following parts: the bioreactor, a water bath, and a measurement system for environmental parameters setting. Figure 1 shows a diagram of its connection array. The bioreactor provided a closed system for adequate cell growth and had a top grip that could be connected to a universal testing machine. The bioreactor was placed inside a water bath. The water bath kept the bioreactor at a constant temperature of (37 ± 1) °C. Temperature was kept constant by circulating warm water with a peristaltic pump at a rate of 120 r.p.m. The system for environmental parameters kept 5% CO_2_ and high humidity flowing into the bioreactor using a micropump. The bioreactor and the water bath were CAD designed using software package (SolidWorks Corp., Waltham, MA, USA) and then fabricated at a local workshop.

### 2.3. Fabrication of the Scaffold and Its Assembly on the In-Vitro Study System

The microfilament was fabricated using PCL in an extruder (Filabot Ex2) at 80 °C equipped with a 1 mm diameter die. Then, the microfilament was stretched until plastic yield, ε ~ 97%. The average diameter of the fabricated microfilament was 90.69 ± 6.37 µm, as measured by optical microscopy. To assemble the scaffold, the microfilaments were arranged as a parallel group of 61 microfilaments by organizing them between the knobs of the grips (see Figure 2). In a previous study made by the authors, the details of the fabrication procedure and mechanical properties of the microfilaments are reported [41]. Briefly, the average diameter of the fabricated microfilaments was 90 µm. The Young modulus of the scaffold was 2184 MPa, and the tensile yield stress was 275 Mpa. In addition, this study showed that PCL microfilaments kept their mechanical properties for 5.3 × 10^5^ cycles under constant loading [41].

The functionality of the biomechanical stimulation system was evaluated by contrasting the cell proliferation inside the study system to that of a scaffold inside a commercial incubator. The scaffold inside the commercial incubator was considered a positive control. Two sets of grips were used: the mobile grip that was placed as part of the study system (see Figure 2a) and the control grip that was kept inside a cell incubator (see Figure 2b). The initial length of the scaffold was 17 mm for the mobile grip and for the control grip. After the scaffold was assembled on the grips, they were sterilized using 25 kGy of gamma rays in a An Ob-Servo Ignis with a Co-60 source gamma irradiator (Izotop, Budapest, Hungary) [42]. For each replica, a new scaffold was assembled and sterilized prior to its use.

### 2.4. Collagen Coating of the Microfilament Scaffold

The mobile grip was placed in a horizontal position on an aluminum holder with a 30 mm petri dish (see Figure 3a). The static grip was directly placed on a petri dish (see Figure 3b). Both grips were placed in such a way that the microfilaments were facing down to ensure the scaffold was in contact with cell culture medium. Once in this position, the scaffolds were coated with collagen I. Approximately 800 µL of collagen was deposited directly on the surface of the microfilaments. The coated scaffolds were then incubated at 37 °C for 30 min inside a cell culture incubator. After 30 min, the remaining collagen was removed by aspiration, and the scaffolds were washed three times with sterile PBS.

### 2.5. Cell Seeding and Biomechanica Stimulation System Arrangement

Cell seeding was made using the configuration showed in Figure 3. Approximately 4 × 10^5^ C2C12 cells suspended in 200 µL of cell growth medium were directly seeded on the scaffolds by adding a drop of cell suspension on the microfilaments. The scaffolds were incubated for 1.5–2 h inside a cell culture incubator to promote cell attachment. Then, 3 mL of cell culture medium was slowly added to the petri dish and scaffolds were kept inside an incubator for 24 h.

The mobile grip with the cell-seeded scaffold was placed inside the bioreactor and the bioreactor was placed inside the water bath (see Figure 4a). The control cell-seeded scaffold was maintained in the petri dish and placed inside a cell culture incubator.

### 2.6. Cell Viability Measurements

Cell proliferation was evaluated by measuring cell viability every 24 h after cell seeding. Table 1 shows the experimental timeline of the viability measurements made. Viability estimation was made by first removing cell growth medium from the cell-seeded scaffolds. Then, 3 mL of solution of cell growth medium with 10% of AlamarBlue^TM^ was prepared and subsequently added to the cell-seeded scaffolds. The scaffolds were incubated at standard conditions (37 °C and 5% CO_2_) for 1 h. Then, the incubated solution was collected and transferred into an optically clear flat bottom microplate. The reduced compound from the AlamarBlue^TM^ reagent was measured with relative fluorescence units (RFU) using a Fluostar Optima (BMG LaBTech, San Diego, CT, USA) microplate reader at 544 nm/590 nm (Ex-Em) wavelength. This test was made three times for statistical analysis and viability was reported as the mean ± standard deviation.

### 2.7. Biomechanical Stimulation of a Cell-Seeded Scaffold

A mechanical stimulation test was used to probe a cell-seeded microfilament scaffold under loading conditions for 24 h. Two samples were used, one was placed inside the in-vitro system and the other sample was placed on a petri dish inside an incubator.

The loading pattern applied to the cell-seeded scaffolds is shown in Figure 5. The scaffold was loaded until the sample reached an elongation of 1 mm. After 1 h of constant displacement, cyclical loading was applied by using a sinusoidal wave function with a frequency of 0.1 Hz and an amplitude of 1 mm. The scaffold was stressed for five cycles, followed by load relaxation at 1 mm of elongation. The displacement plateau remained at 1 mm for 2 h and then this procedure was repeated 10 times. After 10 stimulation cycles, the cells on the microfilament and the control sample were fixed and a staining procedure was followed.

### 2.8. Cell Morphology on the Microfilament Scaffold

Cells were observed on the microfilament scaffolds using fluorescence microscopy. The fixation of cells and staining were performed using the aluminum holding mechanism and a petri dish shown in Figure 3. Scaffolds were washed with PBS and soaked in 4% paraformaldehyde solution for 20 min., then they were washed three times with PBS. After washing the scaffolds, they were incubated in 1% BSA/PBS for 20 min. to permeate the cells; 0.1% of Triton X-100 in PBS was added for 5 min.

Next, 10 µL of phalloidin was diluted in 200 µL of PBS, added to the scaffolds, and incubated for 20 min. Then, DAPI dye was diluted at a ratio of 1:1000 in PBS and added to the scaffold and incubated for 5 min. The cells were rinsed three times with PBS. Once the cells were stained, the scaffolds were removed from the grips and placed individually in a petri dish for microscopy observation. A Leica DMi8 inverted fluorescence microscope equipped with a FITC (Fluorescein isothiocyanate) LP (long pass) filter and DAPI LP filter was used for visualization of the scaffolds. ImageJ software was used to edit the micrographs and obtain a merge of both DAPI and Phalloidin dyes. At the end of the experimental procedure, the cell nuclei number was estimated on each fluorescence micrograph by using ImageJ 2.9.0/1.53t software [43]. The background was removed and then a binary threshold applied. Particles were counted with an area that varied from 50 μm^2^ to 600 μm^2^ and circularity from 0.2 to 0.8.

## 3. Results

### 3.1. Microfilament Scaffold and Assembly of the System

Figure 6 shows different sections of the assembled study system. In (a), the bioreactor is shown. In (b), the bioreactor was placed inside the water bath. The study system held the cell-seeded microfilament scaffold on a vertical position. An augmentation of the bioreactor is shown in (c), where the mobile grip with the microfilament scaffold is appreciated. Figure 6d shows the static adapter in the petri dish. This scaffold remained in a horizontal position in a cell incubator throughout the tests.

### 3.2. Cell Viability Results on the Microfilament Scaffolds

Cell proliferation was assessed as the viability percentage (%) shown in Equation (1).
(1)$$\mathrm{viability}\;\left(\%\right)=\frac{{\mathrm{RFU}}_{\mathrm{scaffolds}}}{{\mathrm{RFU}}_{24\;h}},$$

RFUscaffolds were the fluorescent units of the scaffold at different times and RFU_24 h_ were the control fluorescent units. Viability indicated cells proliferated in the scaffold inside both environments (see Figure 7). For test A1 (24 h, see Table 1), control sample had (100 ± 24)% and (129 ± 71)% in mobile grip. At 48 h, test A2, cell viability of both scaffolds decreased slightly. The change in cell proliferation was attributed to an adjustment of cells to their new environment. At 72 h, test A3 (Table 1), cell viability of the mobile grip scaffold was (276 ± 149)% and the control sample had (295 ± 147)%. Finally, at 96 h, test A4, cell growth increased further. The mobile grip scaffold reached (394 ± 146)%, and viability was (456 ± 59)% in the control scaffold.

Cell growth on both scaffolds showed the regular proliferation rate of C2C12 cells [44]. Cell viability from the mobile grip and control grip were contrasted and results showed control scaffold had slightly higher viability through the assay; however, there were no significant statistical differences between the two, for 24 h, *p* = 0.987; for 48 h, *p* = 0.402; for 72 h, *p* = 0.624; and for 96 h, *p* = 0.103.

As seen in Figure 6c, the mobile grip held the microfilaments vertically aligned inside the system. Results from this test showed that the variation in scaffold orientation did not have a negative effect on cell growth, as there was not a significant change in cell proliferation. For test A3 and A4 (see Table 1), cells grew in both scaffolds with the same growth tendency. Therefore, cell proliferation was independent of the orientation of the microfilament scaffolds. They could be placed in a horizontal position (control scaffold) or in a vertical position (mobile grip scaffold) and cells would colonize and reproduce on the scaffold.

### 3.3. Cell Morphology on the Scaffolds

DAPI/phalloidin staining confirmed presence of cells on the microfilaments. This procedure was used to contrast whether there was a difference between cell morphology on the microfilaments in the grip system and that of the control microfilament sample. Besides, the visualization of the cells on the scaffold could be used to verify the results of the viability assay.

Figure 8a shows the fluorescence micrograph of cells grown on the microfilaments located at the mobile grips, while Figure 8b shows cells on the control scaffold. Both micrographs demonstrated presence of cells. Complete nuclei were distinguished as round blue circles on the surface of a microfilament. The estimated nuclei count in Figure 8a,b was 150 and 110, respectively. A white arrow inside the figure indicates where the microfilaments’ limits are. This result confirmed that cells remained attached to and distributed on the surface of the scaffolds. Some cells stretched across the microfilaments, making a slight connection between the microfilaments.

### 3.4. Cell Morphology of the Biomechanically Stimulated Scaffolds

Figure 9a shows the fluorescence micrographs after 24 h mechanical stimulation of a sample scaffold, and (b) shows the control scaffold. Cell nuclei were 20 in Figure 9a and 82 in Figure 9b. Results confirmed cells remained attached to the scaffold through the loading procedure. Further studies should be conducted to study mechanical stimulation parameters in order to find the best possible outcome for cells.

## 4. Discussion

This work used PCL as raw material for scaffold fabrication because it is biocompatible, has appropriate mechanical properties, and is susceptible to be molded into different shapes [15,17]. Cylindrical, long structures, namely the microfilaments, were chosen as they imitated the myofibers that compose native skeletal muscles. Scaffolds for tissue engineering applications have similar mechanical, physical, and biological properties to those of native tissue [10,11]. These were fabricated by extrusion using a solvent-free method that reduced costs and kept PCL intrinsic properties, as it reduced addition of cell-toxic substances [41].

Microfilaments were arranged as three-dimensional scaffolds on the grip system. This structure replicated the spatial organization of skeletal muscle by organizing the scaffold as parallelly grouped microfilaments [45]. In addition, the scaffold arrangement allowed similar culture conditions to those to in-vivo conditions [29,46]. The fabrication method, in conjunction with the microfilament organization in the grips, allowed this scaffold to benefit from the intrinsic properties of PCL and benefit from using a 3D scaffold. In contrast, PCL is often produced by electrospinning to produce 2D meshes [23,26,29,47].

The grips were designed to hold the microfilaments as a bundle or aggregate of microfilaments and, therefore, utilize their mechanical properties as a group. This was an important attribute, since skeletal muscular tissue, in nature, reacts to force as an aggregate. The mechanical properties of skeletal muscle are the result of grouped fiber behavior. Therefore, these properties cannot be inferred from individual fibers [13,48].

The mobile grip system and the static grips allowed to immobilize the scaffold for surface coating. Since collagen can directly influence cell adhesion, growth, and proliferation, it was applied as a coating to the PCL scaffold to enhance its physicochemical and biological properties [29,32,49]. C2C12 cells bind to the tripeptide Arg-Gly-Asp present in collagen, increasing PCL bioactivity [50]. Cell seeding was made on the immobilized scaffold as myoblasts are anchorage-dependent cells; therefore, they require an adequate surface to begin their growth and proliferate [51].

Viability measurements showed PCL microfilaments comprised a suitable scaffold for cell proliferation, since, throughout the study, cell viability increased (see Figure 7). In addition, the collagen-coated microfilament scaffold confirmed its ability to support cell adhesion and its function as a colonization surface. This was also confirmed with the actin from cell cytoskeleton that was visible directly following the surface of the microfilament, and several scattered nuclei are appreciated on the scaffold in Figure 8. The surface area covered by the nuclei was estimated as a density; nuclei number per microfilament area yielded 13% and 21% in the in-vitro system and control scaffold, respectively. These results confirmed that the biomechanical study system complied with the characteristics that bioreactors require for three-dimensional in-vitro cell culture. Bioreactors provide the optimal environment for cell growth phases, as they provide different conditions for tissue or cells. For instance, Frleta et at. [52] proved that cultivation of diatoms in a controlled bioreactor progressed faster than in an Erlenmeyer flask. In our study, both cell cultures grew in a similar way.

As a result, the system provided suitable conditions for cell survival, growth, and proliferation. The main advantage was the possibility to simulate and modify in-vivo cell growth conditions. For this study, conditions were kept as C2C12 cells required; however, the system allowed for higher or lower CO_2_ % concentration tests to be performed. Also, the system provided a temperature range for cell growth testing. Bioreactors in tissue engineering have proved essential for the development of three-dimensional cell cultures—the chosen conditions depend on the requirements of the investigated tissue [35,36,53].

Mechanical stimulation applied to cells grown on scaffolds has proven to be effective in the fabrication of skeletal muscle tissue engineering structures [1,54,55]. Several loading patterns have been tested in the literature to increase cell proliferation, promote cell differentiation, and cell orientation along the longitudinal axis of the scaffold [2,6,45,56,57,58]. In this study, a simple loading pattern was used to assess whether cells remained attached to the scaffold after 24 h. This test was carried out to probe the functionality of the bioreactor; however, the designed system is capable of performing more complex analysis. Results show cells on the surface of the filament. Their morphology seemed extended, mononucleated, and flat, indicating that they were healthy (see Figure 9). The nuclei number per microfilament area was 9% for the mechanically stimulated scaffold and 28% for the control scaffold. This test showed very promising results, as it confirmed that the biomechanical stimulation system was able to provide an in-vitro environment for inoculated microfilament scaffold.

Mechanical stimulation directs cell physiological changes more quickly than those presented by growth factors; therefore, future research will use different stimulation patterns on the microfilaments. Stimulation patterns include variables such as frequency, intensity, force, and loading time [59]. For example, low stimulation frequencies can induce differentiation, while very strong stimuli can fatally damage cells [60]. In addition, future studies may include evaluation of the biological state of the cell by studying actin cytoskeleton and its associated proteins [38].

## 5. Conclusions

This study demonstrated that the collagen-coated PCL microfilament scaffold provided a suitable template for adhesion and proliferation of C2C12 cells. Cells were uniformly distributed along the surface of the microfilaments. The scaffold had a biocompatible surface that facilitated cellular recognition and, consequently, cellular adhesion. Also, scaffold had a 3D configuration that resembled skeletal muscle, properly mimicking its biological structure. The biomechanical stimulation system was proved to work as an in-vitro test cell growth, indicating that it provided suitable environmental conditions such as the required temperature and CO_2_. These characteristics ensured the survival, growth, and proliferation of the myoblasts on the scaffolds. Overall, the tissue-engineered PCL-based scaffold was found to be a promising model for the study of skeletal muscle and the study system proved promising as a bioreactor for mechanical stimulation of scaffolds.

## 6. Patents

A patent for the biomechanical stimulation system is currently under evaluation. No 278368 (IN2021563599), with classification C12N 58, A6L 27/38.

## Figures and Tables

**Figure 1 polymers-14-05427-f001:**
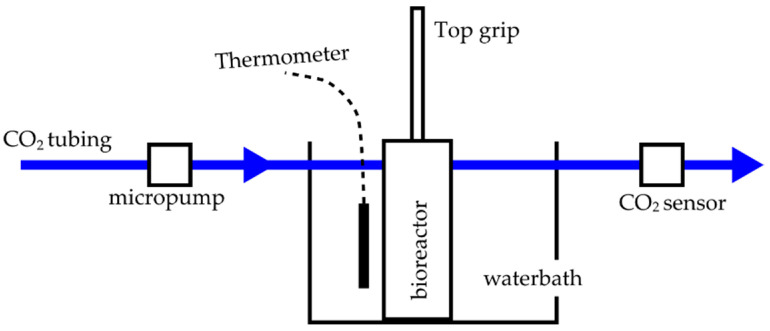
Diagram and connection array of the biomechanical stimulation system.

**Figure 2 polymers-14-05427-f002:**
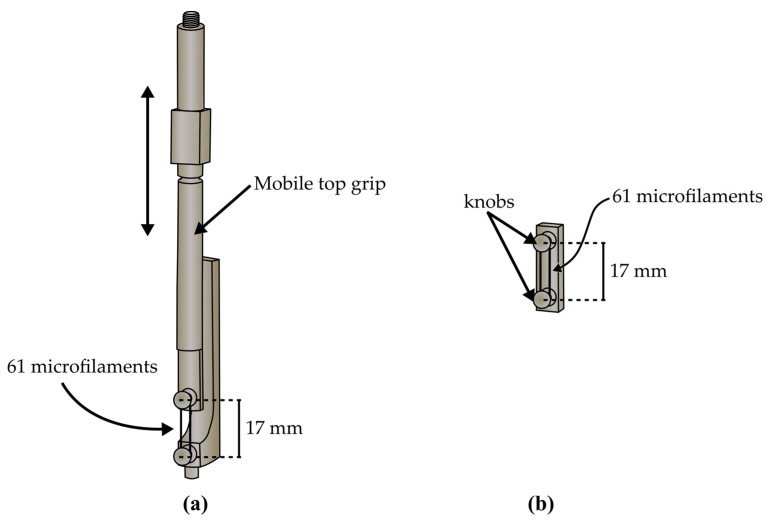
Diagram of grips with microfilament scaffolds. (**a**) shows the mobile grip system and (**b**) shows the control grip.

**Figure 3 polymers-14-05427-f003:**
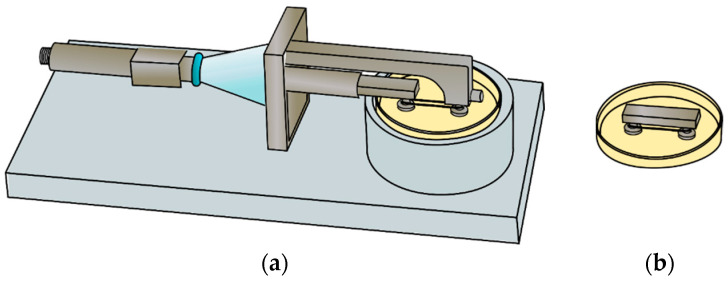
Diagram for collagen coating and cell seeding of the scaffold. (**a**) Mobile grip in aluminum base-holder and in (**b**) static grip in petri dish.

**Figure 4 polymers-14-05427-f004:**
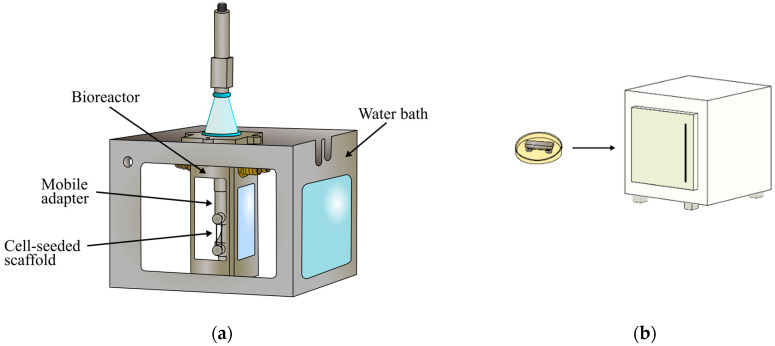
(**a**) Diagram of bioreactor inside water bath. (**b**) Static cell-seeded scaffold to be placed inside the incubator.

**Figure 5 polymers-14-05427-f005:**
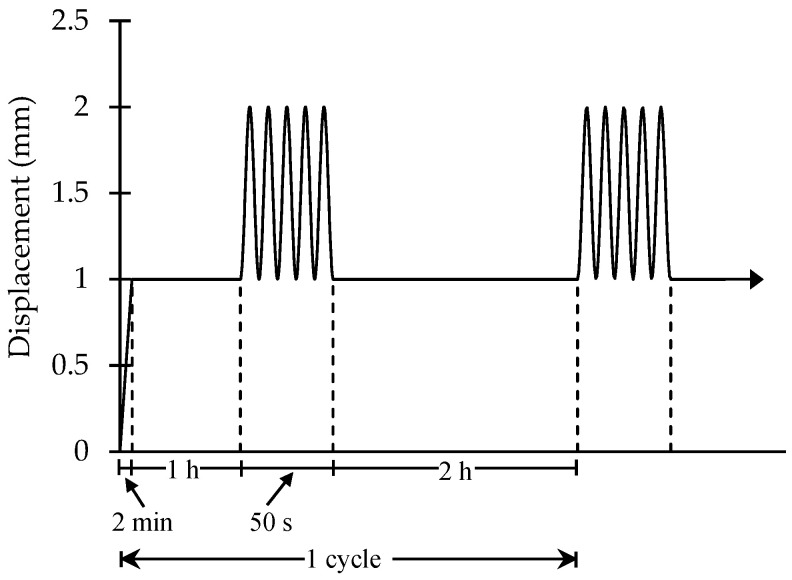
Loading pattern used on cell-seeded scaffold.

**Figure 6 polymers-14-05427-f006:**
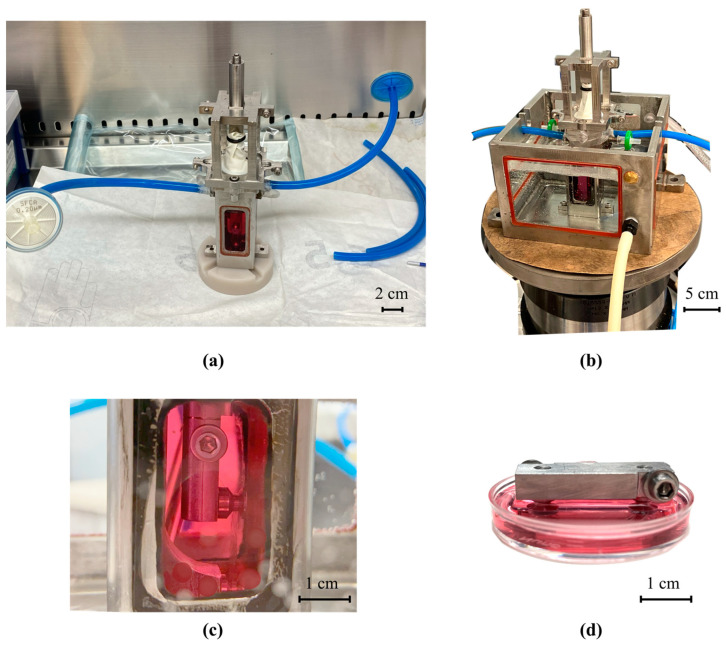
(**a**) Closed bioreactor with input and output circulation hose; (**b**) in-vitro study system; (**c**) scaffold in the mobile grip inside the bioreactor; (**d**) static grip with the scaffold.

**Figure 7 polymers-14-05427-f007:**
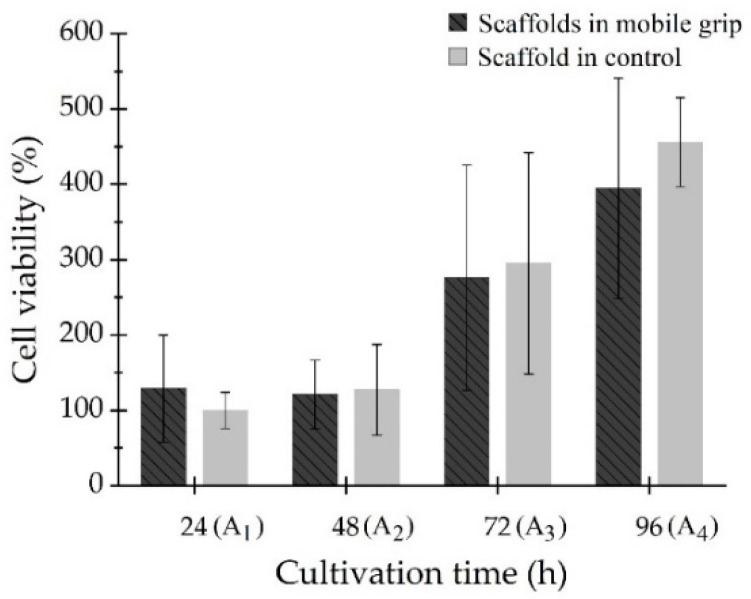
Viability of cells seeded on the collagen-coated PCL scaffolds.

**Figure 8 polymers-14-05427-f008:**
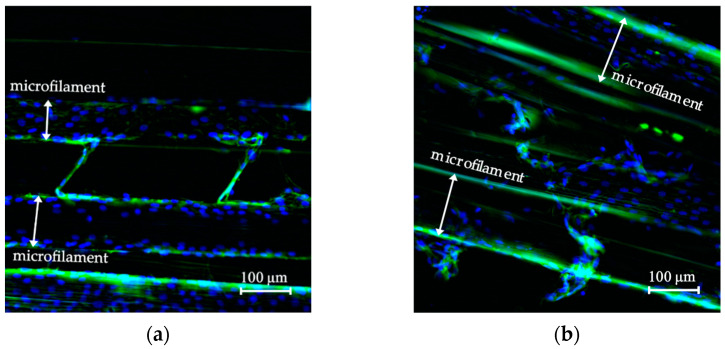
(**a**) Scaffold inside the in-vitro study system without mechanical stimulation and (**b**) control scaffold inside the incubator. Fluorescence micrographs used 20×. Actin filaments of the cytoskeleton are green, and nuclei are blue.

**Figure 9 polymers-14-05427-f009:**
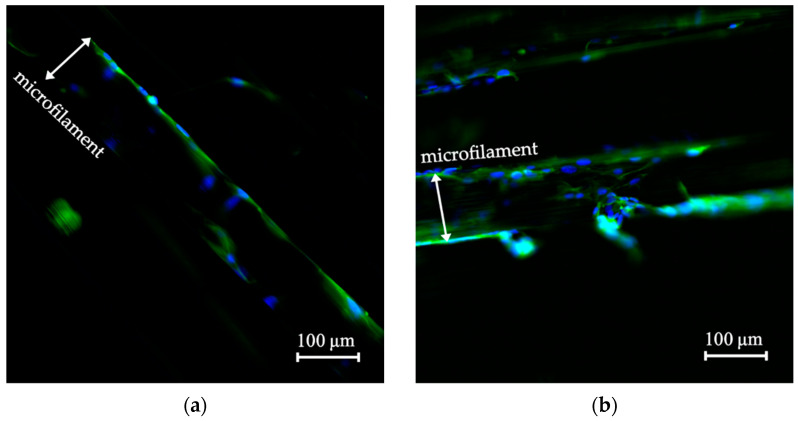
(**a**) Microfilament after 24 h of mechanical stimulation and (**b**) microfilament without mechanical stimulation. Both micrographs were taken at 20× where actin filaments of the cytoskeleton are visible in green, and nuclei are visible in blue.

**Table 1 polymers-14-05427-t001:** Experimental procedure followed on the microfilament scaffolds.

Time (h)	Methodology	Details	Test
0	Cell seeding C2C12	Aluminum holder in incubator	A0
24	Viability measurements	Mobile grip and control grip inside incubator	A1
48	Viability measurements		A2
72	Viability measurements	Mobile grip located in biomechanical stimulation system and control grip inside incubator. Cell density estimation	A3
96	Viability measurements		A4

## Data Availability

The data that support the findings of this study are available upon reasonable request from the corresponding author.

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
