# Peer review of "Muscle-like Scaffolds for Biomechanical Stimulation in a Custom-Built Bioreactor"

_polymers, 2022, doi:10.3390/polym14245427_

Round 1

Reviewer 1 Report

Rojas-Rojas et al develop a novel bioreactor for biomechanical stimulus of muscle-like scaffold or in-vitro bio-system. The number of cell attachment and proliferation on these scaffolds based on PCL were studied by authors. In the end, the bioreactor was found promising for biomechanical applications. The paper is consistent with interesting experimental approach and their presentation. Few revision is necessary in order to improve the paper to be suitable for readership of polymers –

[1] The authors claim various time about the word “biomechanical” system. In addition, the biological tests were performed adequately. However, the mechanical properties were not performed. So, the authors are encouraged to perform mechanical tests on scaffold under compressive or tensile strain.  Mechanical properties like hardness, stiffness, modulus, tensile strength are often used in tissue engineering applications. Such properties are required because the tissue system undergoes mechanical motions under different mechanical strains.

[2] Authors are encouraged to study cell morphology through SEM at higher resolution than fluorescence microscope. The effect of cell seeds after different time could be supported further with SEMs.

[3] Moreover, please refer 4-5 papers from MDPI on subject of the current work in introduction and discuss the advancement of this work over existing work in literature.

Good Luck for revisions !

Author Response

Dear reviewer,

Thank you for reading and reviewing our manuscript. Your suggestions are very valuable to us. You may find we addressed each of your comments. The details are inside the attached document.

Reviewer 2 Report

I'm very glad to become acquainted with such interesting research.

The article is well written and the presented research is well organized. I can only advise to quantify the results of fluorescence micrograph. E.g., nuclei can be counted, microfilament width can be compared etc.

Author Response

Dear reviewer,

Thank you for reading and reviewing our manuscript. Your suggestions are very valuable to us. I will address your comments.

For future studies, we will quantify fluorescence results. However, for this study, it is unfeasible to proceed as suggested. We are still working on the technique for visualization and counting of cells’ nuclei.

We consider this manuscript encloses very valuable results of an interdisciplinary study, however, organizing this information is challenging. As you commented, we will pursue improving our results and communication skills.

The conclusion was revised as the reviewer suggested and minor changes were made for clarification.

Round 2

Reviewer 1 Report

Accept in present form

Author Response

Dear reviewer
Thank you kindly for your acceptance.